# Short-Term Periodized Programming May Improve Strength, Power, Jump Kinetics, and Sprint Efficiency in Soccer

**DOI:** 10.3390/jfmk6020045

**Published:** 2021-05-24

**Authors:** Ai Ishida, S. Kyle Travis, Michael H. Stone

**Affiliations:** 1Department of Sport, Exercise, Recreation, and Kinesiology, East Tennessee State University, Johnson City, TN 37604, USA; travissk@mail.etsu.edu (S.K.T.); STONEM@mail.etsu.edu (M.H.S.); 2Center of Excellence for Sport Science and Coach Education, East Tennessee State University, Johnson City, TN 37604, USA

**Keywords:** collegiate athletes, athlete monitoring, body mass, speed

## Abstract

The purpose of this study was to examine if short-term periodized programming may improve strength, power, jump kinetics, and sprint efficiency in soccer. Seventeen players (19.6 ± 1.6 yrs; 73.8 ± 8.2 kg; 1.77 ± 0.6 m) were divided into two groups based on mean isometric midthigh pull peak force (IPF) (stronger and weaker) and squat jump (SJ) peak power (PP) (higher power and lower power). Eight weaker players were included in the lower power group, while six stronger players were included in the higher power group. Block periodization was adopted to design strength training consisting of 3-week strength endurance and 4-week maximum strength blocks. Performance data included SJ with polyvinyl chloride pipe (SJ0), 20 kgs bar (SJ20), and 40 kgs (SJ40) bar and 20 m sprint across three time points (baseline: T_B_; post-block 1: T_1_; post-block 2: T_2_). Stronger group showed significant increases from T_B_ to T_2_ in SJ20 peak power (PP), net impulse, and allometrically-scaled PP (*p* = 0.005 to 0.01, ES = 0.32 to 0.49). Weaker group demonstrated moderate to large increases from T_B_ to T_2_ in SJ20, allometrically-scaled peak force (PF), PP, and allometrically-scaled PP (*p* = <0.001 to 0.04, ES = 1.41 to 1.74). Lower power group showed significant increases from T_B_ to T_2_ in SJ20 allometrically-scaled PF, net impulse, PP, and allometrically-scaled PP (*p* = <0.001 to 0.026, ES = 1.06 to 2.01). Weaker and less powerful soccer players can benefit from strength-focused training to improve loaded SJ kinetics associating with force production.

## 1. Introduction

Soccer is one of the most popular sports in the world and involves intermittent physical activity such as sprinting, walking, jogging, jumping, kicking, and heading. Two teams consisting of 11 players each play for 90 min on a field that is 90–120 m long and 45–90 m wide [1]. Players often possess high lean body mass, muscle strength and power in the lower extremity, and these capabilities allow the players to produce high running velocities and jump height [2,3,4,5]. For example, Radzimiński et al. [6] reported that a large inverse relationship was observed between fat mass percentage and match-sprinting velocity (*p* < 0.001, r = −0.57). At the collegiate level, soccer players are typically required to play 1 to 2 matches per week for 12 to 24 consecutive weeks. The competitive season schedule would not provide sufficient time to improve physical capabilities. Therefore, obtaining a high level of physical fitness characteristics during the off-season may aid in maintaining physical performance during a competitive season.

Due to a long strenuous competitive season, soccer players would only have sufficient time to develop physical capabilities during the off-season. In collegiate soccer, the off-season lasts for 10 to 12 weeks, so sports scientists and strength and conditioning coaches should select an appropriate training model to improve physical performance during the limited time. Based on current literature [7,8,9,10], block periodization may provide superior fitness adaptations leading to improved sports performance compared to traditional periodization within a limited time frame ranging from 8 to 12 weeks. Block periodization provides concentrated loads (i.e., one summated microcycle) to enhance targeted fitness characteristics (e.g., maximum strength, power) in a training block, and the programmed loading may potentiate the next concentrated loads in the sequential training phase, theoretically [11]. Therefore, block periodization could potentially maximize the long-term effects of training adaptation. When appropriately programmed, block periodization also allows peak performance to be reached at appropriate times [8,9,11]. Considering the collegiate soccer schedule, block periodization may be appropriate when targeting specific training effects in preparation for in-season physical demands and time constraints. However, evidence also indicates that training status can influence gains in physical performance characteristics [12,13,14,15]. Evidence [12,14] indicates that athletes with higher initial maximum strength levels have been shown to gain maximum strength at a slower rate (10–15%) compared to weaker athletes. Thus, it may be beneficial for sport scientists and strength and conditioning coaches to understand the effects of short-term block-style programming on performance changes based on training status.

Isometric midthigh pull (IMTP) is a common performance test to measure maximum isometric strength among various sports, including soccer players [16,17,18,19]. The test is a relatively safe and time-efficient strength assessment [19]. It has been shown that IMTP testing results have been trivial to strongly associate with sprinting, jumping, and agility performance in athletes [16,19,20]. For example, large correlations have been observed between isometric peak force (IPF) and 5 m (*p* ≤ 0.05, r = −0.63) and 30 m print time (*p* ≤ 0.05, r = −0.60) in male sprinters. Kuki et al. [18] showed that 10 m sprint time was faster in collegiate soccer players with higher force capability than those with lower force capability (*p* < 0.05, ES = 1.26). Current evidence [21,22,23] also indicates that IPF largely correlates to one repetition maximum back squat (r = 0.72–0.97), which is considered as a gold standard to assess lower-limb maximum strength. Therefore, IMTP may be a useful tool to assess lower body maximum isometric strength levels in soccer players.

The squat jump (SJ) is also a useful performance test to assess lower-limb power. The SJ test is relatively quick, non-fatiguing, and easy to administer [24]. To perform a SJ, an athlete holds a squat position at a 90° knee angle and vertically jumps up with a concentric movement only without performing a countermovement (i.e., dipping down to propel the body upwards). SJ also shows a very large to perfect correlation between unloaded SJ peak power (PP) and the one repetition maximum back squat (r = 0.70–0.84) [17,25]. Considering these relationships, it would be logical to consider using unloaded SJ PP, which may be useful in determining training program progress in soccer players.

Due to the long strenuous competitive season, it is quite important to know if soccer players can improve physical fitness characteristics in limited periods. However, it is unclear whether short-term periodized programming will be effective for soccer players to improve sprinting and jumping efficiency. Additionally, no data are available in soccer players as to how IPF and power level affect the changes in response to short-term periodized programming, although IMTP and SJ are often used to evaluate maximum isometric strength and peak power in soccer and other team sports [16,17,18,19]. Therefore, the purpose of this study was to investigate the effects of short-term periodized programming on strength, power, jump kinetics, and sprint efficiency in soccer players. It is hypothesized that initial maximum strength and power levels will affect the changes in unloaded and loaded squat jump kinetics and sprinting performance.

## 2. Materials and Methods

### 2.1. Subjects

Seventeen National Association of Intercollegiate Athletics collegiate male soccer players were included in this study (age, 19.6 ± 1.6 yrs; body mass, 73.8 ± 8.2 kg; height, 1.77 ± 0.6 m; training experience, 1 to 4 yrs). Players were excluded from this study if (a) players missed a performance test, (b) they could not complete all weight training sessions because of injuries (non-weight-room), and/or (c) they did not maintain accurate training logs in the weight training sessions. Seven players were excluded due to the absence of the performance tests. All players provided and signed a written informed consent prior to participation. This study was approved by the Institutional Review Board (IRB) at East Tennessee State University (IRB number: c0420.3sdw).

### 2.2. Procedure

This study used a descriptive observational design to monitor jumping and sprinting performance changes in relation to 7 weeks of strength training as a part of an ongoing athlete monitoring program. During the 7 weeks, the jumping and sprinting performance were collected at three points: prior to the start of the first training block (T_B_), the start of the second training block (T_1_), and the end of the second training block (T_2_). Additionally, IMTP was only measured at T_B_ to assess isometric maximum strength. The testing order of strength, power, and sprint assessments were implemented in conjunction with previously reported protocols from our laboratory per Hornsby et al. (2017). After the assessment at T_B_, players were categorized into 2 groups based on IPF from an IMTP: stronger (*n* = 8, training age = 2.0 ± 1.2 yrs) and weaker (*n* = 9, training age −1.8 ± 0.8 yrs) and 2 groups based on PP from unloaded SJ (higher power (*n* = 7, training age = 2.1 ± 1.3 yrs)) and (lower power (*n* = 10, training age = 1.6 ± 0.7 yrs)). The cutoff of IPF (3400 N) was chosen based on the mean PF data from previous literature in youth soccer (17, 21) and other team sports (9, 32). The cutoff of PP was the mean PP of the group (4126 W). Eight weaker players were included in the lower power group, while 6 stronger players were included in the higher power group. During the 7 weeks of the data collection, players completed 2 training sessions in the 1st week and 5 sessions per week during the 2nd to 7th weeks on the practice field (session ratings of perceived exertion = 368 ± 228 in arbitrary units (Table 1)). The strength training program was composed of two blocks of 3 weeks (block 1: strength endurance; block 2: muscle strength), followed by 1 week of a reduced training week.

In this study, block periodization was chosen to design strength training, which consisted of a 3-week accumulation block (high volume, strength endurance emphasis) and 4 weeks of a transmutation block (moderate volume, maximum strength emphasis), with the 4th week serving as a reduced training volume week. Table 2 and Table 3 describe the overall training program over the 7 weeks. Volume load (VL) was calculated by multiplying load (kg) × reps × sets. National Strength and Conditioning Association-certified strength and conditioning coaches observed all the strength training sessions.

The squat jump was measured in a lab setting to assess neuromuscular performance. Prior to jump performance tests, hydration status was evaluated using a refractometer (ATAGO, Tokyo, Japan) to minimize the effect of hydration status on jump and sprint performance. If urine-specific gravity was ≤1.020, players were considered hydrated. If the urine-specific gravity was ≥1.020, players drank water until urine-specific gravity indicated an adequate hydrated state. After the hydration test, a standardized warm-up was completed consisting of 25 jumping jacks and 1 set and 3 sets of 5 dynamic midthigh pull with a 20 kg and 60 kg bar, respectively. After the standardized warm-up, a SJ warm-up was performed with a polyvinyl chloride pipe (SJ0), 20 kg bar (SJ20), and 40 kg (SJ40) bar at 50% and 75% of their perceived maximum efforts before making maximum efforts with each load. Each load was placed on their shoulders. A tester instructed players to stand still on dual force plates (0.91 m × 0.91 m; Rice Lake Weighing Systems, Rice Lake, WI, USA) and to hold a squat position at a 90° knee angle measured with a goniometer. The players vertically jumped from the squat position without a countermovement on the command of “3, 2, 1, jump!”. The trial was removed if a tester visually observed evidence of a countermovement from the force-time curve, the trial was removed, and an additional trial was performed. If the difference of the best two trials was more than 0.2 m in jump height from flight time (JH; m), an additional trial was performed. The best 2 trials in JH were analyzed using a customized program (LabVIEW 2018 Version, National Instruments Co., Austin, TX, USA). JH, peak force (PF; N), allometrically-scaled PF (N·kg^−0.67^), net impulse (N), peak power (PP; W), allometrically-scaled PP (PPa; W·kg^−0.67^), and peak velocity (m·s^−1^) were used as variables of interest. SJ0 PP was used to categorize players into two groups (cutoff = 4126 W). In this study, test-retest reliability was moderate to excellent with an acceptable coefficient of variations (CV) in JH, PF, allometrically-scaled PF, net impulse, PP, and allometrically-scaled PP (intraclass correlation coefficient (ICC): 0.63 to 0.98; CV: 1.3 to 10.1%. peak velocity in SJ had a poor to moderate ICC (ICC: 0.49 to 0.67) with acceptable CV (6.9 to 8.4%) [26,27].

After the SJ test, IMTP testing was conducted at T_B_ on dual force plates (Rice Lake Systems, Rice Lake, WI, USA; 1000 Hz sampling rate). Players were instructed to flex knee joints to 125 ± 5° measured by a goniometer and to maintain an upright torso with extended elbows. Two submaximal IMTP warm-up trials at 50% and 75% of their perceived maximal efforts were performed before completing maximal effort trials. The players pulled upward as fast and hard as possible on the commands of “3, 2, 1, pull!”. If a tester observed a countermovement in the trial (<200 N), the trial was eliminated and repeated. An additional IMTP was performed if the IPF difference of the two trials was more than 200 N. IPF was used as the variable of interest to categorize players into two groups based on isometric strength capabilities. The CV between the best two trails was 5.0% for IPF. IPF in this study showed an excellent correlation with low CV (ICC = 0.94; CV = 5.0%).

A 20 m sprint test was performed (splits at 10 and 20 m; ST10 and ST20) using electronic wireless dual eye timing gates (Timing Ireland, Malahide, Dublin, Ireland) to measure soccer-specific sprinting performance. The height of the timing gates at the start and 10 and 20 m were at a knee joint and hip joint, respectively. Players stood with a staggered stance 0.3 m behind from the start line. The players sprinted two times at their maximal efforts with 2–3 min rest between each sprint. Prior to the two sprint trials, players completed a standardized dynamic warm-up and two prints at 50% and 75% of their perceived maximum efforts. The sprint trial was repeated if the player lost their balance and touched the ground or slowed down prematurely due to a slip. The fastest sprint times (s) in 10 and 20 m sprints were used for data analysis. The study showed lowCVs between the two sprints in 10 and 20 m sprints (10 m: CV= 3.2%; 20 m: CV = 2.2%).

### 2.3. Statistical Analyses

The statistical software R Studio (version 1.1.463, Boston, MA, USA) with the packages dplyr (0.8.5), rstatix (0.4.0), and stats (3.5.3) was used to identify changes in performances within each group across the three testing time points. A two-way analysis of variance was performed to examine the mean difference between groups and three testing time points (T_B_, T_1_, and T_2_). A post-hoc test with a Bonferroni correction was conducted. Effect size (ES) was calculated to identify the magnitude of differences between groups and testing time points. Magnitudes of the ES are interpreted as ES < 0.2= trivial, 0.2–0.6 = small, 0.6–1.2 = moderate, 1.2–2.0 = large, and >2.0 = very large (11). Statistical significance for the analysis was set at *p* ≤ 0.05. All data were expressed as means and standard deviations (SD).

## 3. Results

In the stronger group, statistically significant small to moderate changes were observed between T_B_ and T_2_ in PP (*p* = 0.012, ES = 0.35), net impulse (*p* = 0.029, ES = 0.49), allometrically-scaled PP (*p* = 0.031, ES = 0.32), and peak velocity (*p* = 0.05, ES = 0.46). SJ40 JH also statistically improved from T_B_ and T_2_ in the stronger group (*p* = 0.005, ES = 0.46). In the weaker group, statistically significant moderate to large increases were observed from T_B_ and T_2_ in net impulse (*p* = 0.005, ES = 1.04), allometrically-scaled PF (*p* = 0.041, ES = 1.65), PP (*p* < 0.001 ES = 1.41), PPa (*p* < 0.001, ES = 1.74), and peak velocity (*p* = 0.009, ES = 1.57) (Table 4). In the weaker group, SJ40 JH statistically improved by 0.2 m (*p* = 0.002, ES = 0.99), respectively. However, no statistically significant changes were found in sprint performance between the two groups (*p* > 0.05) (Figure 1).

Interestingly, significant moderate to very large differences were observed between the stronger and weaker groups for SJ0, SJ20, and SJ40 JH (*p* < 0.001, ES = 1.86 to 2.87), net impulse (*p* < 0.05, ES = 1.86 to 2.87), PF (*p* < 0.001, ES = 2.11 to 2.57), allometrically-scaled PF (*p* = <0.001 to 0.01, ES = 1.37 to 2.20), PP (*p*= 0.001 to 0.003, ES = 1.67 to 2.48), and PPa (*p* = 0.002 to 0.05, ES = 1.00 to 1.81) at T_B_, T_1_, and T_2_. However, no significant differences were observed between the groups for SJ0 peak velocity (*p* = 0.07) and for 10 m (*p* = 0.42) and 20 m (*p* = 0.34) sprint time at T_2_.

In the lower power group, statistically significant increases were noted from T_B_ to T_2_ for SJ20 allometrically-scaled PF (*p* = 0.026, ES = 1.06), net impulse (*p* = 0.002, ES = 1.64), PP (*p* < 0.001, ES = 1.06), PPa (*p* < 0.001, ES = 2.01), and peak velocity (*p* = 0.004, ES = 1.46). In the lower power group SJ40, JH and allometrically-scaled PF also significantly increased from T_B_ to T_2_ (JH: *p* < 0.001, ES = 1.88; allometrically-scaled PF: *p* = 0.011, ES = 1.22) (Figure 2). Additionally, significant increases were observed from T_B_ to T_2_ for the lower power SJ0 PP (*p* = 0.02, ES = 1.46) and PPa (*p* = 0.02, ES = 0.98). However, no significant changes were observed in SJ0, SJ20, and SJ40 in the higher power group (*p* > 0.05) (Table 5).

Significant moderate to very large differences were observed between the higher and lower power groups for SJ0, SJ20, and SJ40 JH (ES = 1.35 to 2.23), net impulse (ES = 1.13 to 2.74), and PP (ES = 0.96 to 2.74) at T_B_, T_1_, and T_2_. However, no significant differences were observed between the groups for 10 and 20 m sprint and SJ0 allometrically-scaled PF (*p* = 0.054), PPa (*p* = 0.083), and peak velocity (*p* = 0.28).

## 4. Discussion

The purpose of this study was to examine the effects of a 7-week short-term periodized program on strength, power, jump kinetics, and sprint efficiency in soccer players. The main findings of this study were (a) both stronger and weaker groups significantly improved net impulse, PP, PPa, and peak velocity in SJ20 from T_B_ to T_2_, (b) the weaker group showed a larger magnitude of change in SJ 20 compared to the stronger group, and (c) the lower power group significantly improved force and power output in the unloaded and loaded condition at T_2_, while no statistically significant changes were seen in the higher power group. This study indicates that an initial maximum strength power level would elicit different loaded and unloaded kinetic adaptations. The results of this study also indicate that the strength-emphasized training block is beneficial for the weaker and lower power athletes to improve loaded jump performance.

In both groups, net impulse, PP, PPa, and peak velocity statistically improved from T_B_ to T_2_ at SJ20 and SJ40 in response to 7 weeks of structured strength training. Similar to this study, there is strong evidence indicating that weaker players may make a more substantial improvements in maximum strength measures, jumping, and sprinting in response to emphasizing strength training [12,13,14,28]. For example, Ahtiainen et al. [12] reported that untrained men significantly improved isometric bilateral knee leg extension by 20.9% after 21-week strength training, while the improvement was 3.9% in the trained men. Indeed, current literature [12,13,14,29] has shown that heavy strength-emphasized training programs lead to greater training adaptations associated with force production capacity in weaker athletes. In this study, the weaker group showed greater net impulse changes at SJ20, which may contribute to power production adaptations. In the stronger group, on the other hand, strength-speed or power emphasized training may be a more effective training regimen to improve physical performance [28]. Therefore, based on our findings and previous literature [12,13,14,28,29], strength-emphasized training allows for greater physical performance changes associating with force production changes in weaker athletes.

Interestingly, SJ20 PP in both groups was statistically higher at T_B_ than T_2_. Importantly a large increase was observed in allometrically-scaled PF from T_B_ to T_2_ in the weaker group, while the stronger group did not statistically improve allometrically-scaled PF. These findings indicate that the initial maximum strength level can influence the adaptations in mechanisms resulting in improve SJ20 PP versus increased maximum strength. In the strong group, net impulse and peak velocity were statistically higher at T_2_ compared to T_B_ without significant changes in allometrically-scaled PF. This suggests that PP changes in the stronger group at T_2_ could be explained by an increased average force production capacity at a relatively fast velocity. Similar to our study, James et al. [28] showed that strength-trained men improved net impulse and peak velocity without a significant change in PF after 10 weeks of weightlifting-emphasized training. On the contrary, it seems that the weaker players had a greater trainability to improve mechanical characteristics in all SJ kinematic variables such as net impulse, allometrically-scaled PF, and peak velocity regardless of training emphasis. This is supported by the evidence from Cormie et al. [13], finding that heavy strength training produced substantial improvements in power and movement velocity in relatively weak subjects. Although the magnitude of training adaptations vary, the evidence from this study and previous literature [13,28] suggests that a short-term strength power training program may be beneficial regardless of strength and power level. However, a strength-emphasized training program could produce greater training adaptations in loaded SJ performance in the weaker and less powerful groups than the stronger and more powerful groups. Thus, weaker and less powerful athletes may show greater loaded performance improvement training when their strength levels improve. However, in the stronger and more powerful athletes, training program prescription should be carefully structured and considered for players in an appropriate manner based on initial maximum strength and power levels (e.g., power and plyometric emphasized training for the stronger, more powerful groups).

In soccer, jumping is a fundamental physical performance and often directly involves heading the ball to clear, pass, score, and ability to change direction. Additionally, jump performance is associated with sprinting performance and the ability in the change of directions [30]. Therefore, improving SJ0 performance could be associated with improved soccer-related performance. However, SJ0 performance was not statistically different across the testing three times in stronger or weaker groups. The performance changes might not occur due to lower transfer of training effects to SJ0 compared to SJ20 and SJ40. The training regimen in this study emphasized strength endurance for 3-weeks and muscle strength for 3-weeks. So, selected lower-limb exercises (i.e., back squat and clean pull) were performed at a relatively slower velocity with relatively heavy loads. Thus, the transfer of training effects for SJ0 from the training regimen would be expected to be lower than SJ20 and SJ40 [10,29,31,32,33]. There is strong evidence indicating that mechanical specificity has a profound impact on the magnitude of jump performance changes [8,34]. Similar to our findings, McBride et al. [32] showed that jump performance changes in unloaded and loaded conditions selectively responded to training with various loads in strength power training. Therefore, it seems that mechanical specificity needs to be considered to maximize jump performance changes when a strength training program is designed. SJ0 performance alternations would be expected if a strength-speed emphasis training occurred after the muscle strength-emphasis block [7].

Interestingly, this study suggests that the initial SJ0 PP level affected the adaptations in unloaded loaded jump performance in response to 7 weeks of strength training. The players in the lower power group demonstrated moderate to large changes in SJ0 and SJ20 PP and allometrically-scaled PP at T_2_. The larger adaptation in the lower power group could be explained by the relationship between maximal strength and maximal power; stronger athletes tend to produce greater force and power [15,35,36]. When athletes produce greater forces over a given time, a greater acceleration occurs. In turn, greater force production results in increased power production [15]. This suggests that athletes with lower power would not have higher force production capacity. As a result, strength-emphasized training appears to be more effective for the lower power group than the higher power group. Measuring PP from SJ0 in conjunction with maximum strength may be a useful tool to determine an effective training regimen.

There are four main limitations in this study. First, the sample population is male collegiate soccer players. The finding of this study may not apply to other populations in soccer (e.g., female players, professional male players, etc.). Second, there is no consensus as to the cutoff of IPF and unloaded SJ PP in order to group soccer players. Thirdly, morphological (i.e., muscle fiber type and muscle cross-sectional area) and neuromuscular (i.e., muscle electromyography) changes were not evaluated in this short-term study. Finally, the detailed training background of individual players was not measured in this study (e.g., type of resistance training). Therefore, the player’s resistance training background may affect the results of this study. Future studies should measure morphological data with a larger sample size at different competition levels.

## 5. Conclusions

Our study indicated that short-term periodized programming may be effective in improving unloaded and loaded jump kinetics in weaker and lower power soccer players. Based on SJ kinetic data, it also should be noted that, with a maximum strength emphasis, there appears to be a degree of specificity for the gains in power and movement velocity. Weaker and less powerful soccer players can benefit from a strength-focused training regimen to improve loaded SJ compared to the stronger and more powerful players. Measuring force production and power levels are recommended to aid in assessing the effect of strength training on jumping performance changes.

## Figures and Tables

**Figure 1 jfmk-06-00045-f001:**
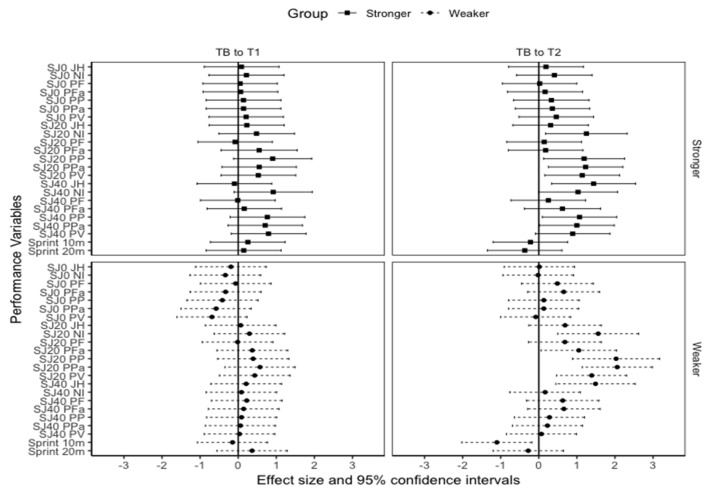
Effect size and 95% confidence intervals in squat jump and sprint changes in the stronger and weaker groups. Note. TB = Baseline. T1 = The start of the second training block. T2 = The end of the second training block. SJ0 = Squat jump with a polyvinyl chloride pipe. SJ20 = Squat jump with 20 kgs bar. SJ40 = Squat jump with bar loaded to 40 kgs. JH = Jump height. NI = Net impulse. PF = Peak force. Pfa = Allometrically-scaled peak force. PP = Peak power. PPa = Allometrically-scaled peak power. PV = Peak velocity.

**Figure 2 jfmk-06-00045-f002:**
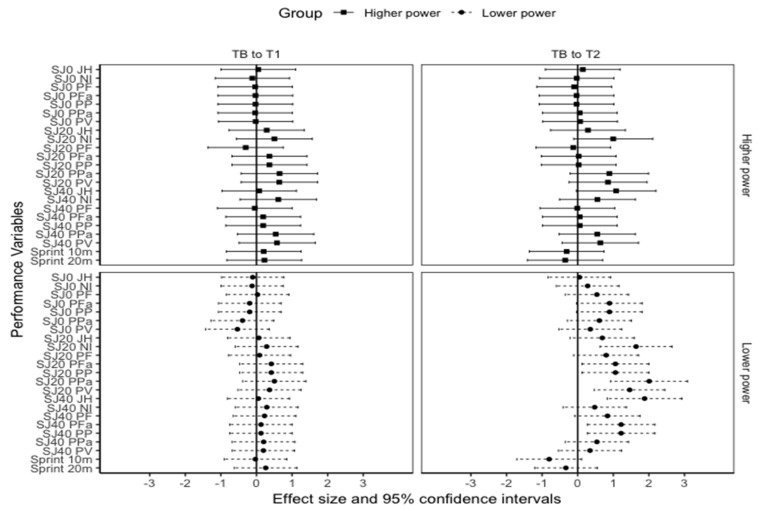
Effect size and 95% confidence intervals in squat jump and sprint changes in the higher and lower power groups. TB = Baseline. T1 = The start of the second training block. T2 = The end of the second training block. SJ0 = Squat jump with a polyvinyl chloride pipe. SJ20 = Squat jump with 20 kgs bar. SJ40 = Squat jump with bar loaded to 40 kgs. JH = Jump height. NI = Net impulse. PF = Peak force. Pfa = Allometrically-scaled peak force. PP = Peak power. PPa = Allometrically-scaled peak power. PV = Peak velocity.

**Table 1 jfmk-06-00045-t001:** Overall schedule for 7 weeks.

Week	Monday	Tuesday	Wednesday	Thursday	Friday	Saturday	Sunday
1	PT, Wt	OFF	Wt	OFF	Wt	OFF	OFF
2, 3	Tr, Wt	Tr	Tr, Wt	Tr	Tr, Wt	OFF	OFF
4	PT, Tr, Wt	Tr	Tr, Wt	Tr	Tr, Wt	OFF	OFF
5, 6	Tr, Wt	Tr	Tr, Wt	Tr	Tr, Wt	Match	OFF
7	Wt	Tr	Wt	Tr	PT, Wt	OFF	OFF

Note: Wt = Weight training. Tr = Soccer-related training sessions. PT = Performance testing.

**Table 2 jfmk-06-00045-t002:** Strength training program for 7 weeks.

Phase	Week	Sets × Reps	Daily Relative Intensities (D1, D2, D3)	Volume Load (Kgs)
SE	1	3 × 10 (3 × 5)	ML, ML, VL	17,221 ± 2548
SE	2	3 × 10 (3 × 5)	M, M, L	18,800 ± 2735
SE	3	3 × 10 (3 × 5)	H, H, L	19,992 ± 2966
MS	4	5 × 5 (5 × 3)	ML, ML, VL	16,333 ± 2145
MS	5	3 × 5 (3 × 3)	M, M, L	12,159 ± 1486
MS	6	3 × 5 (3 × 3)	H, H, L	12,664 ± 1670
MS	7	3 × 3	VL, VL, VL	8053 ± 1595

Note: D1 = Day 1. D2 = Day 2. D3 = Day 3. SE = Strength endurance. MS = Maximum strength. VL = Very light (65–70%). L = Light (70–75%). ML = Moderate light (75–80%). M = Moderate (80–85%). MH = Moderate heavy (85–90%). H = Heavy (90–95%). (3 × 5, 5 × 3 and 3 × 3) = denotes sets and reps for clean pull and pull from knee.

**Table 3 jfmk-06-00045-t003:** Exercise selection for strength training programs.

Day	Strength Endurance	Muscle Strength
Day 1 and Day3	Back Squat	Back Squat
	BB Shoulder Press	BB CG Push Press
	DB Lunge	DB Step Up
	BB Bench Press	BB Bench Press
	DB Triceps Extension	
Day 2	Clean Pull	Pull from Knee
	Straight-Leg Deadlift	Straight-Leg Deadlift
	Bent-Over Row	Bent-Over Row
	DB Pull Over	DB Pull Over
	DB Biceps Curl	

Note. BB = Barbell. CG = Clean grip. DB = Dumbbell.

**Table 4 jfmk-06-00045-t004:** Changes of selected squat jump and sprint variables in stronger and weaker groups.

Variables	Stronger (*n* = 8)			Weaker (*n* = 9)		
	T_B_	T_1_	T_2_	T_B_	T_1_	T_2_
SJ0						
JH (m)	0.37 ± 0.06	0.38 ± 0.05	0.38 ± 0.05	0.28 ± 0.03 §	0.28 ± 0.04 §	0.28 ± 0.04 §
NI (N∙s^−1^)	216.2 ± 16.4	219.7 ± 15.9	223.9 ± 20.5	177.3 ± 24.7 §	170.0 ± 18.7 §	176.9 ± 21.9 §
PF (N)	1903.3 ± 164.0	1912.4 ± 192.2	1907.0 ± 152.6	1516.1 ± 157.5 §	1506.9 ± 114.0 §	1588.2 ± 137.2 §
PFa (N∙kg^−0.67^)	101.8 ± 7.8	102.3 ± 83.2	103.0 ± 7.4	89.9 ± 6.5 §	88.1 ± 3.7 §	93.9 ± 5.9 §
PP (W)	4693.5 ± 62.5	4783.9 ± 673.4	4929.4 ± 791.9	3621.9 ± 601.7 §	3408.8 ± 402.4 §	3696.2 ± 523.8 §
PPa (W∙kg^−0.67^)	251.5 ± 37.5	257.3 ± 44.8	267.9 ± 53.1	214.7 ± 31.9 §	199.3 ± 20.0 §	218.8 ± 31.0 §
PV (m∙s^−1^)	2.9 ± 0.2	3.0 ± 0.3	3.1 ± 0.4	2.8 ± 0.3	2.7 ± 0.2 §	2.8 ± 0.3
SJ20						
JH (m)	0.28 ± 0.05	0.28 ± 0.05	0.28 ± 0.03	0.20 ± 0.03 §	0.20 ± 0.02 §	0.21 ± 0.02 §
NI (N∙s^−1^)	232.4 ± 19.9	236.9 ± 19.5	242.6 ± 22.1 *	184.1 ± 16.3 §	189.4 ± 24.1 §	202.4 ± 19.0 *¶§
PF (N)	2038.6 ± 186.2	2033.1 ± 189.3	2049.4 ± 173.3	1652.0 ± 122.0 §	1650.2 ± 98.1 §	1740.2 ± 114.5 ¶§
PFa (N∙kg^−0.67^)	93.9 ± 7.4	94.7 ± 7.3	94.4 ± 6.5	82.2 ± 3.8 §	83.7 ± 3.1 §	87.3 ± 2.5 *¶§
PP (W)	4519.4 ± 625.3	4596.2 ± 584.9 *	4742.2 ± 687.0 *	3340.7 ± 321.0 §	3514.5 ± 582.2 §	3823.4 ± 363.9 *¶§
PPa (W∙kg^−0.67^)	208.6 ± 30.6	217.8 ± 40.4	218.9 ± 34.0 *	166.1 ± 12.7 §	179.1 ± 28.7 §	192.1 ± 17.1 *§
PV (m∙s^−1^)	2.6 ± 0.2	2.7 ± 0.3	2.7 ± 0.2 *	2.3 ± 0.1 §	2.5 ± 0.3	2.6 ± 0.2 *§
SJ40						
JH (m)	0.20 ± 0.04	0.20 ± 0.03	0.22 ± 0.03 *¶	0.12 ± 0.02 §	0.13 ± 0.01 §	0.14 ± 0.02 *¶§
NI (N∙s^−1^)	238.7 ± 24.2	259.3 ± 32.8	259.1 ± 28.2	192.6 ± 23.8 §	194.7 ± 25.0 §	196.8 ± 27.5 §
PF (N)	2202.7 ± 183.4	2202.0 ± 180.8	2223.1 ± 178.4	1823.8 ± 123.1 §	1843.6 ± 106.7 §	1898.4 ± 109.6 §
PFa (N∙kg^−0.67^)	89.5 ± 5.8	89.9 ± 6.6	90.9 ± 5.2	79.2 ± 3.7 §	79.8 ± 3.3 §	82.0 ± 2.6 §
PP (W)	4316.1 ± 593.3	4703.9 ± 817.7	4728.4 ± 679.5	3323.2 ± 433.3 §	3363.7 ± 458.8 §	3447.0 ± 497.5 §
PPa (W∙kg^−0.67^)	175.5 ± 23.7	192.6 ± 37.4	193.9 ± 30.2	144.4 ± 18.0 §	145.8 ± 20.1	149.3 ± 23.1 §
PV (m∙s^−1^)	2.2 ± 0.2	2.4 ± 0.3	2.4 ± 0.3	2.1 ± 0.2	2.1 ± 0.2	2.1 ± 0.3 §
Sprint						
10 m (s)	1.81 ± 0.12	1.83 ± 0.06	1.79 ± 0.07	1.82 ± 0.06	1.81 ± 0.06	1.76 ± 0.05
20 m (s)	3.07 ± 0.16	3.09 ± 0.09	3.03 ± 0.09	3.09 ± 0.09	3.12 ± 0.07	3.07 ± 0.09

Note. T_B_ = Baseline. T_1_ = The start of the second training block. T_2_ = The end of the second training block. SJ0 = Squat jump with a polyvinyl chloride pipe. SJ20= Squat jump with 20 kgs bar. SJ40 = Squat jump with bar loaded to 40 kgs. JH = Jump height. NI = Net impulse. PF = Peak force. PFa = Allometrically-scaled peak force. PP = Peak power. PPa = Allometrically-scaled peak power. PV = Peak velocity. * = denotes statistically different from baseline (*p* ≤ 0.05). ¶ = denotes statistically different from T_1_ (*p* ≤ 0.05). § = denotes statistically significant from the stronger group.

**Table 5 jfmk-06-00045-t005:** Changes of selected squat jump and sprint variables in higher and lower power groups.

	Higher Power (*n* = 7)			Lower Power (*n* = 10)		
Variables	T_B_	T_1_	T_2_	T_B_	T_1_	T_2_
SJ0						
JH (m)	0.37 ± 0.08	0.37 ± 0.06	0.38 ± 0.06	0.29 ± 0.03 §	0.29 ± 0.04 §	0.29 ± 0.04 §
NI (N∙s^−1^)	222.5 ± 11.4	220.6 ± 19.5	221.9 ± 29.2	176.9 ± 20.6 §	174.3 ± 21.1 §	183 ± 20.0 §
PF (N)	1919.4 ± 178.2	1911.8 ± 241.9	1899.8 ± 211.8	1543.5 ± 166.3 §	1547.8 ± 134.4 §	1625.1 ± 135.0 §
PFa (N∙kg^−0.67^)	102.8 ± 7.6	102.6 ± 10.0	102.6 ± 10.1	90.3 ± 6.3 §	89.3 ± 3.7 §	95.1 ± 4.3 §
PP (W)	4892.1 ± 506.5	4850.6 ± 731.1	4903.8 ± 1011.4	3590.1 ± 471.0 §	3499.6 ± 441.7 §	3837.4 ± 494.9 ¶§
PPa (W∙kg^−0.67^)	263.4 ± 35.9	262.0 ± 46.5	266.5 ± 63.7	210.1 ± 21.8 §	201.9 ± 20.0 §	224.8 ± 26.2 ¶
PV (m∙s^−1^)	3.0 ± 0.3	3.0 ± 0.3	3.0 ± 0.5	2.8 ± 0.2 §	2.7 ± 0.2 §	2.8 ± 0.3 §
SJ20						
JH (m)	0.28 ± 0.06	0.28 ± 0.05	0.28 ± 0.04	0.21 ± 0.03 §	0.21 ± 0.03 §	0.22 ± 0.03 §
NI (N∙s^−1^)	233.2 ± 24.1	238.3 ± 20.6	246.1 ± 20.8	188.3 ± 18.2 §	193.2 ± 25.7 §	203.9 ± 18.8 *¶§
PF (N)	2074.7 ± 168.3	2052.6 ± 205.6	2065.8 ± 188.3	1665.4 ± 122.5 §	1674.8 ± 109.5 §	1759.6 ± 115.8 ¶§
PFa (N∙kg^−0.67^)	95.5 ± 6.5	96.1 ± 6.7	95.5 ± 6.0 §	82.3 ± 3.6 §	83.9 ± 3.1 §	87.3 ± 2.8 *¶§
PP (W)	4594.7 ± 653.6	4806.4 ± 386.9	4864.3 ± 647.4	3405.9 ± 343.7 §	3475.6 ± 456.0 §	3829.8 ± 336.3 *¶§
PPa (W∙kg^−0.67^)	211.7 ± 32.0	223.6 ± 39.9	225.6 ± 32.4	168.1 ± 12.9 §	178.8 ± 27.1 §	190.1 ± 14.7 *§
PV (m∙s^−1^)	2.6 ± 0.2	2.7 ± 0.3	2.7 ± 0.3	2.3 ± 0.1 §	2.5 ± 0.3	2.5 ± 0.2 *
SJ40						
JH (m)	0.20 ± 0.05	0.20 ± 0.05	0.21 ± 0.05	0.13 ± 0.03 §	0.13 ± 0.02 §	0.15 ± 0.03 *¶§
NI (N∙s^−1^)	241.4 ± 25.1	257.9 ± 42.2	251.0 ± 39.8	195.3 ± 23.8 §	202.1 ± 27.3 §	208.7 ± 35.3 §
PF (N)	2240.0 ± 162.1	2237.0 ± 162.6	2238.9 ± 196.9	1835.6 ± 122.0 §	1855.0 ± 107.6 §	1919.7 ± 112.2 ¶§
PFa (N∙kg^−0.67^)	90.8 ± 4.9	91.3 ± 5.5	91.0 ± 3.7	79.3 ± 3.4 §	79.8 ± 3.2 §	82.8 ± 2.4 *¶§
PP (W)	4408.4 ± 573.3	4740.8 ± 944.6	4608.2 ± 918.5	3357.9 ± 425.0 §	3471.9 ± 470.1 §	3659.2 ± 614.3 §
PPa (W∙kg^−0.67^)	178.9 ± 23.4	194.1 ± 42.3	187.6 ± 38.2	145.1 ± 17.2 §	149.4 ± 19.3 §	158.2 ± 27.4
PV (m∙s^−1^)	2.3 ± 0.2	2.4 ± 0.4	2.3 ± 0.3	2.1 ± 0.2	2.1 ± 0.2	2.2 ± 0.3
Sprint						
10 m (s)	1.79 ± 0.12	1.81 ± 0.05	1.76 ± 0.05	1.83 ± 0.15	1.83 ± 0.09	1.78 ± 0.08
20 m (s)	3.04 ± 0.06	3.07 ± 0.06	3 ± 0.07	3.11 ± 0.10	3.13 ± 0.07	3.08 ± 0.09

Note. T_B_ = Baseline. T_1_ = The start of the second training block. T_2_ = The end of the second training block. SJ0 = Squat jump with a polyvinyl chloride pipe. SJ20= Squat jump with 20 kgs bar. SJ40= Squat jump with bar loaded to 40 kgs. JH = Jump height. NI = Net impulse. PF = Peak force. PFa= Allometrically-scaled peak force. PP = Peak power. PPa = Allometrically-scaled peak power. PV = Peak velocity. * = denotes statistically different from baseline (*p* ≤ 0.05). ¶ = denotes statistically different from T1 (*p* ≤ 0.05). § = denotes statistically significant from the stronger group.

## Data Availability

Not applicable.

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
