# Peer review of "Short-Term Periodized Programming May Improve Strength, Power, Jump Kinetics, and Sprint Efficiency in Soccer"

_jfmk, 2021, doi:10.3390/jfmk6020045_

Round 1
Reviewer 1 Report
Abstract, lines 2-5
It seems that the subjects were divided according to the mean isometric-mid- thigh pull peak force and squat jump peak power (higher power and lower power). The groups were having the same strength profile? I mean, the participant who was stronger in thigh pull peak force had also the higher jump performance? I assume that this was the case but this have to be clear in the abstract.
Abstract, Results
The authors should highlight in the abstract that only a block training program was adopted in the present investigation and there was no comparison with a traditional training program.
Abstract, Conclusion (last sentence)
The concluding sentence is very general and does not depict the main finding of the present investigation. The authors should present if the two groups (i.e., well trained vs. not well-trained individuals) were responded differently in the different training interventions.
Materials and Methods, Procedures, lines 1-6
The authors should minimize the abbreviations in this important section, where is explained how the participants were divided into the two groups. Moreover, the abbreviations should be minimized throughout the manuscript. In many cases, it is very difficult to follow the text with all these abbreviations.
Materials and Methods, Procedures
In this section the authors should describe only the procedures (time line of the intervention and the training program). The performance assessments should be presented in a sub-section of the procedures section.
Author Response
A: Thank you for the constructive comments. Please see the responses below.
Abstract, lines 2-5
It seems that the subjects were divided according to the mean isometric-mid- thigh pull peak force and squat jump peak power (higher power and lower power). The groups were having the same strength profile? I mean, the participant who was stronger in thigh pull peak force had also the higher jump performance? I assume that this was the case but this have to be clear in the abstract.
A: I clarified the changes as requested (abstract and line 109 to 110). Stronger athletes are not necessarily allocated to the higher power group.
Abstract, Results
The authors should highlight in the abstract that only a block training program was adopted in the present investigation and there was no comparison with a traditional training program.
A: I clarified in the abstract section
Abstract, Conclusion (last sentence)
The concluding sentence is very general and does not depict the main finding of the present investigation. The authors should present if the two groups (i.e., well trained vs. not well-trained individuals) were responded differently in the different training interventions.
A: I specified the population. Please check it out.
Materials and Methods, Procedures, lines 1-6
The authors should minimize the abbreviations in this important section, where is explained how the participants were divided into the two groups. Moreover, the abbreviations should be minimized throughout the manuscript. In many cases, it is very difficult to follow the text with all these abbreviations.
A: I modified the abbreviations.
Materials and Methods, Procedures
In this section the authors should describe only the procedures (time line of the intervention and the training program). The performance assessments should be presented in a sub-section of the procedures section.
A: I tried to clarify the procedure in Line 107 to 120.
Reviewer 2 Report
The article investigated the effects of 7-weeks of periodized resistance training on strength and power outcomes in male soccer players. The research itself is of interest to the scientific, and strength and conditioning communities. That being said, some areas of the manuscript require further refinement (writing errors and some more substantial points that require consideration). I have provided my specific comments below.
L28: change ‘that’ to ‘and’
L31: Unequal spacing after ‘players’. Also, ‘a high amount of lean body mass’
L32: ‘These capabilities’
L40-42: This sentence could be refined.
L47: Remove the first ‘next’
L55: Suggest placing 10-15% in brackets after slower rate.
L61: trivially
L62: ‘have been observed’
L66: Please check if IPF has been defined earlier.
L70: ‘The squat jump’
L71: ‘easy to administer’
L80: ‘unclear whether’
L96: should this be ‘did not maintain accurate training logs’?
L159: ‘at Tb’ can be removed.
L165: IMPT
L167: ‘trials’
L239: ‘of a 7-week’ and ‘periodized program on’
L263: There appears to be an error in writing here.
L286: perhaps ‘ability to change direction’
L324: ‘short-term’
General comments:
It would be useful to know how many of the low strength athletes were also in the low power group. I can see the group numbers are slightly different, and although can be reasonable assumed that those with low strength also had low power, this should be described or displayed more clearly. Perhaps as a simple figure with connecting lines between individual data for the strength and power measure, and group included in.
L166: Here there is a clear statement about IPF being used to categorize players into a low versus high group. Please include a similar statement about the classification or metric used to determine the low versus high power groups in an appropriate section (apologies if already included and I have missed this).
There is limited presentation of between group differences for the outcome variables despite the stats approach listed on L184. In text the only description that is evident appears on L199-200 for sprint performance. Nor do the figures or tables display between group differences. In my opinion this limits the interpretation of the study and makes some points in the discussion not entirely sound. For example, L242: ‘the weaker group showed the larger magnitude of change in SJ20 than the stronger group. However, this can not be said from the results presented as the group comparison was not presented. Thus, it is unknown if the greater change in SJ20 was statistically greater than the stronger group. Please amend and also check other statements. Further detail about between group statistical outcomes should also be provided in relevant sections.
It would also be useful to know if the players were already performing resistance training prior to the study, especially the stronger athletes. If so, perhaps this can partly explain the generally smaller amount of adaptation seen in the stronger group. On this note, as a similar training program was delivered for each group perhaps the stimulus was less optimal for the stronger group? This last point is not a criticism of the study design but rather just a point for thought.
L281: Perhaps here the previous point could come into context. The authors state that the program should be carefully structured based on strength level but no further thought about what should be considered is provided. I suggest a small expansion of this point to include the authors perspectives about training considerations for different strength levels based on their findings from the study.
Author Response
The article investigated the effects of 7-weeks of periodized resistance training on strength and power outcomes in male soccer players. The research itself is of interest to the scientific, and strength and conditioning communities. That being said, some areas of the manuscript require further refinement (writing errors and some more substantial points that require consideration). I have provided my specific comments below.
A: Thank you for the thoughtful feedback. I edited all the comments as your request.
General comments:
It would be useful to know how many of the low strength athletes were also in the low power group. I can see the group numbers are slightly different, and although can be reasonable assumed that those with low strength also had low power, this should be described or displayed more clearly. Perhaps as a simple figure with connecting lines between individual data for the strength and power measure, and group included in.
A: I will mention the group shift in Line 109.
L166: Here there is a clear statement about IPF being used to categorize players into a low versus high group. Please include a similar statement about the classification or metric used to determine the low versus high power groups in an appropriate section (apologies if already included and I have missed this).
A: I added the sentence in Line 157
There is limited presentation of between group differences for the outcome variables despite the stats approach listed on L184. In text the only description that is evident appears on L199-200 for sprint performance. Nor do the figures or tables display between group differences. In my opinion this limits the interpretation of the study and makes some points in the discussion not entirely sound. For example, L242: ‘the weaker group showed the larger magnitude of change in SJ20 than the stronger group. However, this can not be said from the results presented as the group comparison was not presented. Thus, it is unknown if the greater change in SJ20 was statistically greater than the stronger group. Please amend and also check other statements. Further detail about between group statistical outcomes should also be provided in relevant sections.
A: I added the results between groups in the results section. The sentence in the discussion was described based on the ES, not comparing between the stronger and weaker group. If you look at the ES from Tb to T2 in the same metrics, the weaker group showed greater performance changes.
It would also be useful to know if the players were already performing resistance training prior to the study, especially the stronger athletes. If so, perhaps this can partly explain the generally smaller amount of adaptation seen in the stronger group. On this note, as a similar training program was delivered for each group perhaps the stimulus was less optimal for the stronger group? This last point is not a criticism of the study design but rather just a point for thought.
A: I agree but, unfortunately, we could not track the weight training history of the players.
L281: Perhaps here the previous point could come into context. The authors state that the program should be carefully structured based on strength level but no further thought about what should be considered is provided. I suggest a small expansion of this point to include the authors perspectives about training considerations for different strength levels based on their findings from the study.
A: I added my perspective after Line 285 to 293.
Round 2
Reviewer 1 Report
No further comments
Author Response
N/A
Reviewer 2 Report
L14: 'were included'
L42: This statement still doesn't seem correct as the authors state that there is a long season, but also sufficient time in the off-season. What is the point trying to be made here? The next statement seems like the more likely scenario here.
L11: remove 'the'
L121: 'were included in he the lower power'
L22s: 'for SJ0'
L237: 'for 10m'
It is unclear from the tracked changes were further perspective about the training for low and high power group has been added. Please refer to my comment from previous round.
Acknowledged that resistance training history could not be evaluated. However, this in itself should be acknowledged in the appropriate section of the methods.
Author Response
Thank you for the additional comments. We fixed the manuscript as your request.
L14: 'were included'
A: I fixed
L42: This statement still doesn't seem correct as the authors state that there is a long season, but also sufficient time in the off-season. What is the point trying to be made here? The next statement seems like the more likely scenario here.
A: all the practitioners know that soccer players cannot improve physical performance during the season because they need to play 1 or 2 matches [er week. Instead, the off-season can emphasize physical development because there are no games allocated.
L11: remove 'the'
I could not find "the" in L11
L121: 'were included in he the lower power'
A: I assume you mentioned the L131 and fixed it.
L22s: 'for SJ0'
A: I assume you expect me to change from "in" to "for" in L241-247 and 257 to 261.
L237: 'for 10m'
A: I fixed it
It is unclear from the tracked changes were further perspective about the training for low and high power group has been added. Please refer to my comment from the previous round.
A: I have added the aspects of training prescription on Line 361 to Line 372. I am not sure what information you expect me to add more based on our findings and previous literature.
Acknowledged that resistance training history could not be evaluated. However, this in itself should be acknowledged in the appropriate section of the methods.
A: Please see the Line from 128 to 130.